# The Plutonium Temperature Effect Program

**Nicolas Leclaire \*** and **Vaibhav Jaiswal**

Institut de Radioprotection et de Sûreté Nucléaire (IRSN), PSN-RES/SNC/LN,
F-92260 Fontenay-aux-Roses, France; vaibhav.jaiswal@irsn.fr
**\*** Correspondence: nicolas.leclaire@irsn.fr

**Abstract:** Various theoretical studies have shown that highly diluted plutonium solutions could have a positive temperature effect, but up to now, no experimental program has confirmed this effect. The French Plutonium Temperature Effect Experimental Program (or PU+ in short) aims to effectively show that such a positive temperature effect exists for diluted plutonium solutions. The PU+ experiments were conducted in the "Apparatus B" facility at the CEA VALDUC research center in France. It involved several sub-critical approach-type experiments using plutonium nitrate solutions with concentrations of 14.3, 15, and 20 g/L at temperatures ranging from 20 to 40 °C. Fourteen (five at 20 g/L, four at 15 g/L, and five at 14.3 g/L) phase I experiments (consisting of independent sub-critical approaches) were performed between 2006 and 2007. The impact of the uncertainties on solution acidity and plutonium concentration made it difficult to demonstrate the positive temperature effect, requiring an additional phase II experiment (with a unique plutonium solution) from 22 to 28 °C that was performed in July 2007. This phase II experiment has shown the existence of a positive temperature effect of ~+5.17 pcm/°C (from 22 to 28 °C for a plutonium concentration of 14.3 g/L). It has recently been possible to confirm the results of this program with MORET 5 calculations by generating thermal scattering data $S(\alpha,\beta)$ at the correct experimental temperatures. This paper finally presents a fully documented experimental program highlighting the Plutonium Temperature Effect theoretically described in the literature. Its high level of precision and its "one-step" approach to criticality allowed it to show a significant positive temperature effect for a rather small variation of temperature (+6 °C). The order of magnitude of the effect was confirmed with Monte Carlo calculations using thermal scattering data for hydrogen in the solution produced by IRSN for the purpose of the comparison.

**Keywords:** criticality; plutonium; solution; temperature; thermal scattering law

## 1. Introduction

The plutonium temperature effect (also called PU+ later in the paper) corresponds to an increase in the reactivity of plutonium solution due to an increase in temperature. This effect is particular to plutonium solutions [1] and cannot be observed with uranium solutions, for which a temperature increase will always lead to a negative reactivity effect. The PU+ effect finds its origins in three different physical effects:

- An expansion effect (a decrease in density and an increase in volume versus an increase in temperature),
- a Doppler broadening effect due to the influence of temperature on cross-sections,
- an effect on the thermal scattering of hydrogen in the solution.

The expansion effect can be easily quantified through measurements of solution density. The Doppler effect of low worth is taken into account through proper cross-section processing achieved by using NJOY 2016.35 [2] (MacFarlane and Muir, 1994) to generate JEFF-3.3 cross-section libraries at the appropriate temperatures in the ACE format for the Monte Carlo code MORET 5 [3] that has been used in this work.

Yamamoto and Miyoshi were the first to describe the PU+ effect theoretically [1]. They showed that, depending on the plutonium isotopic vector, there is a plutonium concentration above which the effect is negative and below which the effect becomes positive and tends to increase with the decrease in concentration. Using the four-factor formula and perturbation theory, they analyzed the parameters that had an influence on the PU+ effect. They showed that the temperature coefficient of a solution is positive if the adjoint flux increases with neutron energy between 0.05 eV and 0.2 eV. They also pointed out that $^{241}$Pu tended to increase the temperature coefficient because of the energy dependence of the capture cross-section (see Figure 1). Moreover, since $^{241}$Pu in a plutonium solution decays into $^{241}$Am with time, it reinforces the effect for the same reason. Finally, they highlighted the impact of neutron absorbers such as cadmium, gadolinium, and samarium. Soluble absorbers in a plutonium solution lead to a positive temperature coefficient for higher-concentration plutonium solutions since their capture cross-sections decrease with increasing neutron energy between 0.05 eV and 0.2 eV.

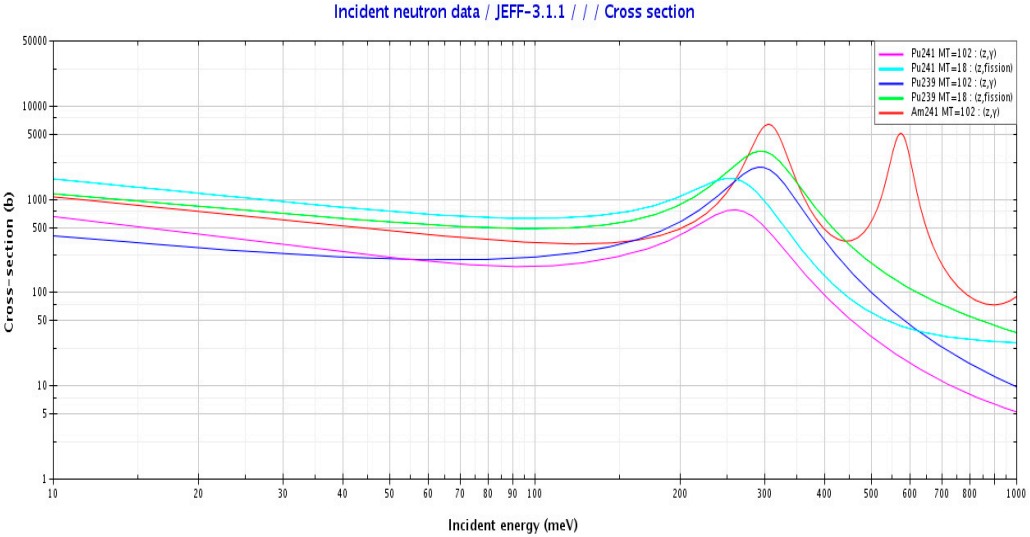

**Figure 1.** Cross-sections of plutonium in the thermal energy range.

Understanding the effect of temperature on plutonium solutions is essential because plutonium solutions are used in many fuel cycle facilities (e.g., reprocessing plants), and a temperature increase in normal or abnormal situations could lead to a criticality risk [4–6]. Moreover, the effect of temperature on Pu solutions is required for neutronics studies of some molten salt reactors. To demonstrate the positive effect of temperature, the French PU+ program was carried out for diluted plutonium solutions. The aim of the program, which was to demonstrate the positive temperature effect experimentally, was reached, and it constituted an opportunity to improve the treatment of S($\alpha$,$\beta$) used by the neutronics codes.

This paper describes the experimental program presented in [5], provides an insight into the amount of uncertainties, and provides a benchmark model on which calculation codes such as the multi-group APOLLO2-MORET 4 from the CRISTAL V1.2 package [7], the APOLLO2-MORET 5 codes from the CRISTAL V2.0 package [8], and the IRSN continuous energy MORET 5 [3] code can be tested. To calculate the configurations correctly at various experimental temperatures, generating thermal scattering data was necessary at the required temperatures. The generation of these data is explained in the paper, and they are used to assess the plutonium temperature effect. It is shown that with thermal scattering data evaluated at the right temperature, a positive effect on $k_{eff}$ is observed, which allows validation of the implementation of S($\alpha$,$\beta$).

## 2. Experimental Installation

The PU+ experimental program was conducted at Apparatus B in the CEA/Valduc research center from 2006 to 2007. The experimental device has been commonly used

for several years to assemble configurations with epithermal and thermal neutron energy spectra. It presents the advantage of being flexible, accommodating configurations with solutions and configurations involving lattices of rods.

*Description of the Set-Up*

The experimental set-up used in the plutonium temperature effect program is given in Figure 2. The experimental core consists of two concentric cylindrical vessels:

- An inner vessel that accommodates the plutonium solutions during the experiment,
- an outer vessel that provides neutron reflection by a water layer of 22 cm both laterally and under the plutonium vessel. It helps maintain a stable temperature for the device.

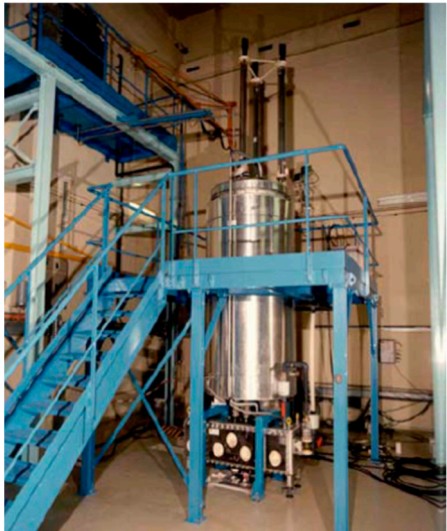

**Figure 2.** Experimental device of the PU+ program. Reflector vessel with instrumentation above.

The inner plutonium vessel is a 0.3 cm thick, 215.5 cm high stainless steel cylindrical vessel (Z2 CND 17-12 on Figure 3) with an inner radius of 37.8 cm (Figure 3). It is closed by a cover, and the bottom is manufactured to allow the complete draining of the solution at the end of the sub-critical approach (Figure 3). The plutonium solution is introduced into the core from two filling and draining pipes connected to the vessel's bottom and sides, respectively. Another pipe located at the top of the vessel returns the plutonium solution directly to the storage tank in the event of an overflow. The vessel ensuring neutron reflection is also a cylindrical stainless steel vessel (Z2 CND 18-10 in Figure 3) with an inner radius of 60.1 cm. This is a 0.3 cm thick, 238.5 cm high vessel surrounded from all sides by 10 cm of rock wool acting as thermal insulation (not visible in Figure 3). The water is introduced from two filling and draining pipes (Figure 4), respectively, connected to the bottom and sides of the vessel. Similar to the plutonium vessel, this vessel also has an overflow pipe at the top.

The covers of the two vessels house three liquid-level measurement devices referred to as limnimeters. These limnimeters (Figure 5) are equipped with two needles. The first needle is used to measure the liquid level with high accuracy (of the order of 0.001 cm). The second needle is a security needle that allows the opening of the drainage valves when a predetermined level of plutonium solution is reached during a sub-critical approach. Two of these limnimeters are used for measuring the level of the plutonium solution. The first one is equipped with an extension and measures the liquid level from the bottom of the vessel up to 130 cm, and the second one measures the liquid level from 100 cm up to the overflowing pipe (205 cm). The third limnimeter is used for measuring the water level in the reflector vessel.

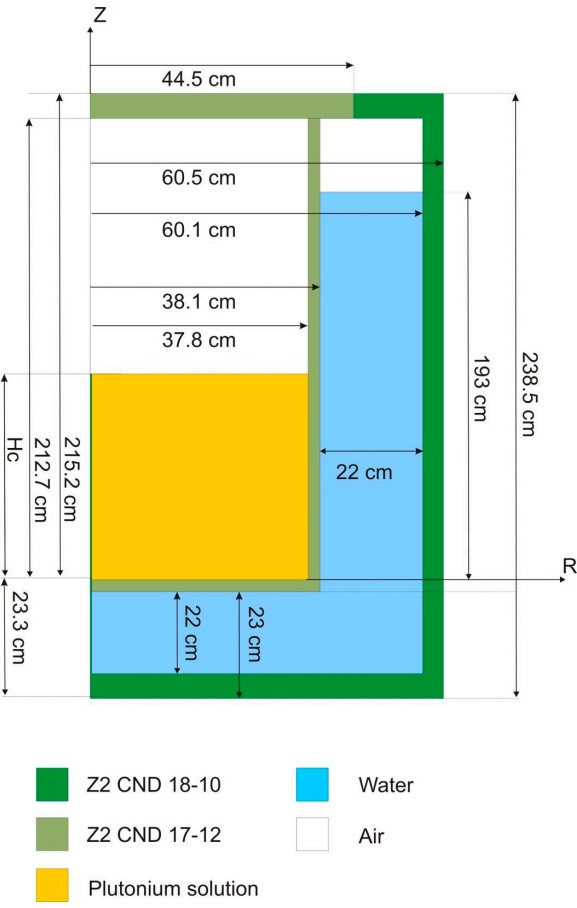

**Figure 3.** Scheme of the benchmark configuration (vertical cut).

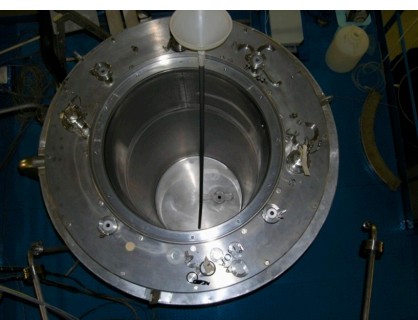

**Figure 4.** Plutonium vessel with the machined zone in the bottom for solution draining.

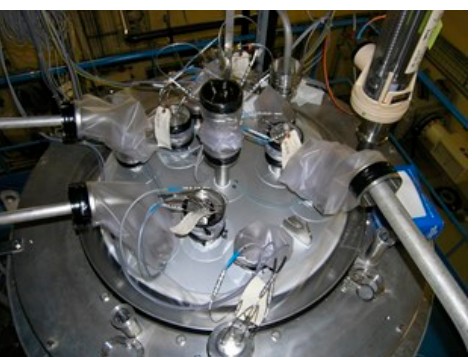

**Figure 5.** Overview of the solution tank with instrumentation.

The experimental set-up is also equipped with six temperature probes (Figure 6) in the plutonium solution and the water reflector. To regulate the temperature of the plutonium solution, the experimental set-up has also been equipped with a heating system. The storage tank for the water reflector (which has a 5000 L capacity) is equipped with six 18 kW electric heaters, a regulator system to maintain the predefined temperature, and a circulation pump to homogenize the temperature in the tank. The plutonium storage tank (enclosed in an insulated containment) is also equipped with a heating device. The solution inside the storage tank is heated by hot air with a temperature of 70 °C directed onto the outer wall of the tank. The device itself consists of a set of 6 kW resistances, a fan coil unit, and a regulator system to maintain the predefined temperature. The previous two heating devices are used to preheat the water reflector and the plutonium solution before they are transferred to the water reflector vessel and the plutonium vessel. All transfer pipes, the water reflector vessel, and the plutonium tank storage have been thermally insulated to limit heat loss. During the course of an experiment, the temperature of the plutonium solution in the plutonium vessel is regulated by water circulation in the reflector (no heating device is in direct contact with the plutonium solution) by using the Vulcatherm. The Vulcatherm itself is equipped with a pump and several 6 kW heating resistances. The experimental set-up has also been equipped with an ultimate safety system that is triggered in the highly unlikely event that the standard safety systems (dropping both the plutonium solution and the water reflector in their respective storage tanks) do not function properly. This ultimate safety system consists of injecting an acidic solution of natural gadolinium into the plutonium solution using two independent pipes connected to the top of the plutonium vessel.

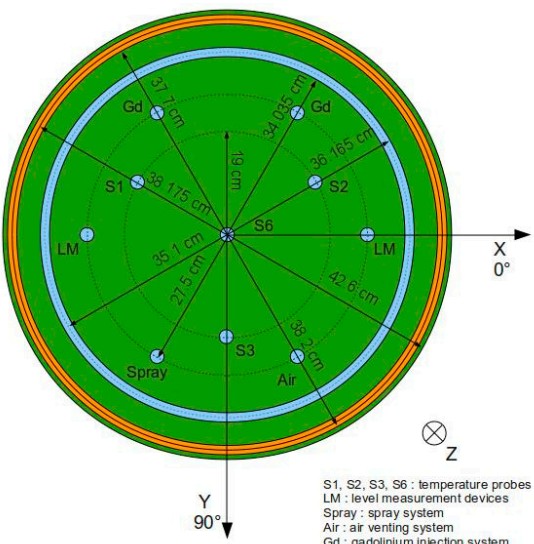

**Figure 6.** Schematic view from above of the solution tank and position of the instrumentation.

## 3. Description of the Program

### 3.1. Experimental Procedure

During the course of the experimental program, several independent sub-critical approach experiments (designated as phase I experiments) were performed for three plutonium concentrations and different temperatures. The aim was to determine the solution level corresponding to $k_{eff}$ = 1 experimentally. This was carried out by progressively increasing the level of the solution by adding small amounts of plutonium solution, thus increasing the reactivity of the system. The steps of the solution level increase were determined based on the prediction of critical heights and according to safety standards linked to the reactivity increase versus solution height. This process was pursued as long as the effective multiplication factor $k_{eff}$ was lower than or equal to $1-\beta_{eff}/10$ (where $\beta_{eff}$ is the effective delayed neutron fraction, estimated to be equal to 210 pcm). Then, the curve giving the inverse of the neutron counting rate versus solution height was extrapolated to

zero (see Figure 7). The obtained value was the critical height of the solution. At the end of each approach, both the plutonium solution and water reflector are transferred to their respective storage tanks.

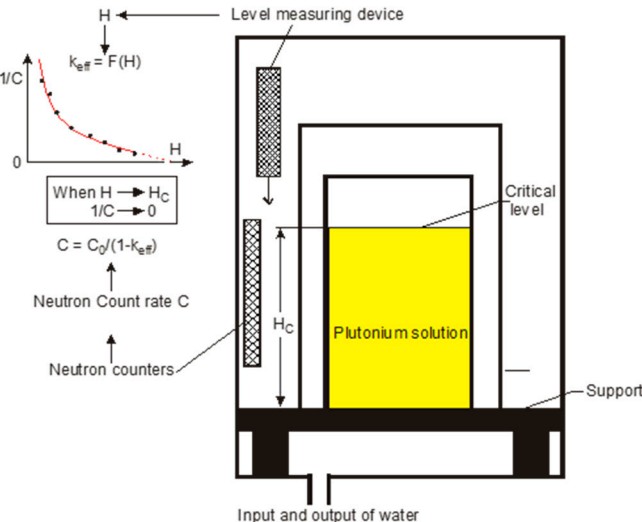

**Figure 7.** Principle of sub-critical approach technique.

### 3.2. Proposed Experiments

The plutonium solutions were prepared at three target concentrations from a mother solution initially prepared in 1993. $^{241}$Am had to be removed from the solution. The acid normality was set to 1N, and the targeted plutonium concentrations were 14.3 g/L, 15 g/L, and 20 g/L (see Table 1). Two phases of experiments were defined:

- Phase I comprised fourteen experiments (five at 20 g/L, five at 15 g/L, and four at 14.3 g/L). Four of these phase I experiments were terminated prematurely due to technical issues. For the 2998, 2999A, 3001A, and 3002A experiments, the approach to criticality was stopped too far away from criticality. In fact, the last solution height was far from the extrapolated height to criticality. Because the uncertainty of extrapolation was large in comparison with the obtained critical height, the experiments were not considered to constitute proper benchmarks by the experimentalists and were therefore discarded from the selection.
- Phase II comprised three experiments (at 14.3 g/L).

**Table 1.** Characteristics of the plutonium solution for the various experiments (the uncertainties given correspond to 2σ).

| Experiment | T [°C] | C(Pu) [g/L] at 21 °C | Dilation Factor | Critical Mass (kg) ± 3σ | H+ (mol/L) at 21 °C | ρ [g/cm³] at 21 °C | Average Extrapolated Critical Height [cm] |
|---|---|---|---|---|---|---|---|
| | | | | Phase 1 | | | |
| 2997 | 22.20 ± 010 | 19.665 ± 0.095 | | 3.43 ± 0.04 | 1.0700 ± 0.011 | 1.06678 ± 0.00004 | 38.869 ± 0.075 |
| 2999B | 39.84 ± 0.07 | 20.146 ± 0.042 | 0.99243 | 3.45 ± 0.04 | 1.0707 ± 0.021 | 1.06787 ± 0.00005 | 38.104 ± 0.086 |
| 3000 | 22.18 ± 0.08 | 20.111 ± 0.044 | | 3.40 ± 0.04 | 1.059 ± 0.041 | 1.06760 ± 0.00003 | 37.630 ± 0.066 |
| 3001B | 40.00 ± 0.07 | 15.001 ± 0.013 | | 5.83 ± 0.04 | 1.032 ± 0.011 | 1.05804 ± 0.00006 | 86.583 ± 0.217 |
| 3002B | 39.97 ± 0.07 | 14.895 ± 0.045 | 0.99241 | 5.80 ± 0.04 | 1.032 ± 0.020 | 1.05798 ± 0.00007 | 86.778 ± 0.181 |
| 3003 | 22.15 ± 0.08 | 15.010 ± 0.098 | | 5.74 ± 0.04 | 1.042 ± 0.005 | 1.05812 ± 0.00003 | 85.191 ± 0.200 |
| 3004 | 22.19 ± 0.09 | 14.246 ± 0.023 | | 8.65 ± 0.04 | 1.034 ± 0.006 | 1.05699 ± 0.00007 | 135.255 ± 0.024 |
| 3005 | 39.99 ± 0.08 | 14.290 ± 0.020 | 0.99238 | 8.48 ± 0.04 | 1.036 ± 0.005 | 1.05707 ± 0.00003 | 132.142 ± 0.030 |
| 3006 | 30.09 ± 0.09 | 14.285 ± 0.036 | | 8.36 ± 0.04 | 1.032 ± 0.004 | 1.05709 ± 0.00001 | 130.374 ± 0.027 |
| 3007A | 22.21 ± 0.09 | 14.294 ± 0.010 | | 8.55 ± 0.04 | 1.033 ± 0.008 | 1.05681 ± 0.00004 | 133.254 ± 0.031 |
| | | | | Phase 2 | | | |
| 3007B | 28.14 ± 0.07 | 14.294 ± 0.010 | 0.99735 | 8.47 ± 0.04 | 1.033 ± 0.008 | 1.05681 ± 0.00004 | 131.997 ± 0.118 |
| 3008 | 28.13 ± 0.08 | 14.277 ± 0.023 | | 8.83 ± 0.04 | 1.039 ± 0.007 | 1.05707 ± 0.00002 | 137.751 ± 0.025 |
| 3009 | 40.01 ± 0.10 | 14.067 ± 0.015 | | 8.63 ± 0.04 | 1.038 ± 0.011 | 1.05648 ± 0.00001 | 136.710 ± 0.032 |

In this table, A refers to room temperature (21 °C) and B to other temperatures (28 °C or 40 °C).

At the beginning of the program (Phase I), only independent sub-critical approaches were planned for safety reasons. In fact, the command control system could not be easily modified to drain some solution from the tank. Since phase I experiments are entirely independent of one another, even if the experimental uncertainties are very low (see Table 2), a potential reactivity effect lower than the level of uncertainties would not be significant in comparison with the uncertainties associated with the two independent approaches. Moreover, it should be noted that the two solutions at 14.3 g/L, even if they are assumed to be identical, have been drained into the storage tank and could have been contaminated differently.

**Table 2.** Experimental uncertainties of the PU+ program.

| Parameter | | Variation in the Calculation | Uncertainty (1σ) | $\Delta k_{eff}(1\sigma) \times 10^5$ |
|---|---|---|---|---|
| Temperature (°C) | Doppler effect | 3 | 0.13 | Negligible |
| | Solution density | | | Negligible |
| | Water density | | | Negligible |
| Acidity (mol/l) | | 0.1 | 0.004 | 5 |
| Plutonium concentration (g/L)—systematic | | 0.1 | 0.033 | 89 |
| Plutonium concentration (g/L)—statistical | | | 0.005 | 14 |
| Plutonium valence | | 100% | 5.77% | 7 |
| Density of solution (g/cm$^3$)—systematic | | 0.01 | 0.00065 | Negligible |
| Density of solution (g/cm$^3$)—statistical | | | 0.00002 | Negligible |
| Am concentration (%) | | 20 | 11.55 | 9 |
| Isotopic Composition $^{240}$Pu (%) (a) | | 0.1 | 0.01 | 9 |
| Isotopic Composition $^{241}$Pu (%) (b) | | 0.1 | 0.0025 | Negligible |
| Tank wall composition (%) | | 1 | 0.58 | 9 |
| Detected impurities (boron) (mg/L) | | 0.44 | 0.25 | Negligible |
| Impurities below the detection limit (mg/L) | | 2 | 0.29 | 79 |
| Solution height (cm) | | 0.24 | 0.05 | 8 |
| Tank radius (cm) | | 0.2 | 0.062 | 17 |
| Tank thickness (cm) | | 0.05 | 0.006 | Negligible |
| TOTAL | | | | 123 |

In 2006, a new technical solution (phase II) was studied in order to make the draining of the solution possible. After approval by the safety authorities, new sub-critical approaches (experiments 3007A and 3007B) were proposed. Contrary to previous experiments, those experiments were perfectly correlated.

In order to highlight a significant PU+ effect with regard to the experimental uncertainties, it was paramount to design the experiments as closely correlated as possible with one another.

The idea was to use the same solution at various temperatures (without draining the plutonium solution from the vessel) as follows:

- First, a standard sub-critical approach at the initial temperature was performed;
- for safety reasons, a small amount of the solution was drained to reduce the plutonium mass because the positive temperature effect would lead to a reactivity increase (roughly 1% of the initial solution will have to be drained);
- the temperature of the solution was slowly increased up to the targeted temperature (either 28 °C or 40 °C) by heating the water reflector, and finally, a standard sub-critical approach was performed at the final temperature.

This idea was used for the design of experiments 3007A and 3007B. Except for the temperature discrepancy, the experiments were fully correlated, which allowed magnifying the effect of the reactivity effect caused by the temperature increase. It is crucial when observing minor effects compared to the experimental uncertainties (see Table 2).

*3.3. Characterization of the Plutonium Solutions*

The plutonium solutions were prepared by nitric dilution of an initial plutonium nitrate solution at 28.86 g/L. The program started with a plutonium solution at 20 g/L and continued with solutions with a decreasing plutonium content (15 and 14.3 g/L). The list of

experiments with the corresponding plutonium concentrations and temperatures is given in Table 1.

The bulk density of the solution was determined through the helium pycnometry technique. Moreover, the solutions were fully characterized through a dosage using the Davis and Townsed method. This method consists of reducing plutonium at valency III with an excess of copper. Then, copper is oxidized by potassium dichromate. A first equivalent point is obtained. After adding a mixture of acids, plutonium is oxidized at valency IV by potassium dichromate. A second equivalent point is obtained. The mass of plutonium in the sample is determined by the mass of potassium dichromate necessary between the first and second equivalent points.

$$C(\text{Pu}) = \frac{(m_1 - m_2) \times T_{\text{Cr}} \times \text{MA}_{\text{Pu}}}{\text{PE}} \times 1000 \tag{1}$$

where:

$m_1$ is the mass of potassium dichromate of the first equivalent point;
$m_2$ is the mass of potassium dichromate of the second equivalent point;
$T_{\text{Cr}}$ is the concentration of potassium dichromate in (N/g);
$\text{MA}_{\text{Pu}}$ is the average atomic mass of plutonium (g/mol);
PE is the mass of the sample;
$C(\text{Pu})$ is given in (g/kg).

The plutonium isotopic composition is reported in Table 3. The impurities were measured using the ICP-MS and ICP-AES techniques (see Table 4).

**Table 3.** Isotopic composition of plutonium.

| Isotope | $^{238}\text{Pu}$ | $^{239}\text{Pu}$ | $^{240}\text{Pu}$ | $^{241}\text{Pu}$ | $^{242}\text{Pu}$ |
|---|---|---|---|---|---|
| Content in % | $0.1940 \pm 0.0050$ | $76.9410 \pm 0.0200$ | $20.6980 \pm 0.0200$ | $1.0800 \pm 0.0050$ | $1.0870 \pm 0.0050$ |

It should be noted that, for ease of use in using the benchmark model, the instrumentation immersed in the solution tank monitoring the temperature and level of the solution was not modeled. The critical height of the solution had to be modified to account for that removal.

**Table 4.** Measured impurities using inductive coupled plasma mass spectrometry (ICP-MS) and inductive coupled plasma absorption and emission mass spectrometry (ICP-AES) for the plutonium solution (19.67 g/L) measured on 2 February 2007.

| Element | Concentration [mg/L] | Element | Concentration [mg/L] |
|---|---|---|---|
| $^{241}\text{Am}$ | 16 | Mg | 2.3 |
| B | 2.2 | Ni | 19.6 |
| Ba | 1.5 | Pb | 0.4 |
| Ca | 4.8 | Th | 2 |
| Cr | 6.7 | Zn | 3.5 |
| Fe | 34 | | |

The elements with content lower than 1 mg/L include Ag, Al, As, Au, Be, Cd, Ce, Co, Cs, Cu, Dy, Er, Eu, Ga, Gd, Ge, Hf, Hg, Ho, Ir, La, Li, Lu, Mn, Mo, Nb, Nd, Os, Pd, Pr, Pt, Rb, Re, Rh, Ru, Sb (not detected), Sc, Se, Sm, Sn (not detected), Sr, Ta, Tb, Te, Ti, Tl, Tm, U, V, W, Y, Yb, and Zr.

### 3.4. Measurement of Temperatures

Six temperature probes (each using three Pt 100 type thermocouples at three different heights along a hollow support tube) were positioned both in the water reflector and in the plutonium solution (four probes). Four probes are distributed axially and radially to cover the different expected three critical heights so that a proper temperature profile in the plutonium solution can be determined. The remaining two probes are distributed axially and radially in the water reflector. Each probe consists of three platinum temperature sensors that have been calibrated using recognized temperature standards.

The measurement uncertainty of the thermocouples is $\pm 0.17\,^\circ\text{C}$ in the operating range of 20–40 $^\circ\text{C}$.

The thermocouples are housed in stainless steel tubes (304L equivalent to Z2 CN 18-10) with an external and internal radius of 0.40 cm and 0.31 cm, respectively. The actual thermocouples are fixed on the outside of the tubes. The plutonium solution or water from the reflector can enter these tubes from the bottom or through holes in the sides where the thermocouples are attached to the tubes.

## 4. Evaluation of Experimental Data

The level of precision (level of rigor) associated with the plutonium temperature effect experiments is a prerequisite to being able to detect the PU+ effect out of the experimental uncertainties. Consequently, sensitivity studies were performed to assess the impact of the various experimental uncertainties on the configuration reactivity in accordance with the recommendations of the ICSBEP uncertainty guide [9]. The 3D APOLLO2-MORET 4 Monte Carlo computations were used to determine the sensitivity of the results to variations in geometrical and material data. The reactivity changes produced by the above tools were adopted as the associated components of the $k_{eff}$ uncertainty.

A specific treatment was applied to impurities in the plutonium nitrate solution. Namely, the detected and measured impurities were modeled while other impurities were omitted, adding some uncertainty.

The various components of the $k_{eff}$ errors are shown for experiment 3007B in Table 2. The overall uncertainty is calculated as the square root of the sum of squares of its individual components. The level of overall uncertainties is quite comparable to other experiments within the range of 0.13% to 0.15%. The main uncertainties subject to variations depending on the case are the uncertainty of the plutonium concentration, the critical height of the solution, and the temperature of the solution.

It can be seen that the uncertainties of the plutonium concentration and those related to the impurities contained in the solution have a paramount effect on the overall uncertainty.

## 5. Analysis of the Experimental Results

### 5.1. Determination of a Benhcmark Model

Within the configurations, some details are of low importance regarding criticality. Therefore, simplifications were proposed with no influence on $k_{eff}$. The calculation models, called "benchmarks", derived from the experiments include simplifications. A sketch of the benchmark model is given in Figure 3. Further calculations of $k_{eff}$ are based on these models. Among the simplifications, there are:

- The removal of impurities from the fissile solution was announced as below a detection limit,
- the omission of neutron counters (Figures 8 and 9),
- the omission of temperature probes and level measurement devices, which are accounted for by a correction of the level of solution,
- the omission of drainage pipes below the reflector tank and the solution tank.

### 5.2. Codes and Associated Libraries

Various codes and libraries were used to calculate the PU+ experiments:

- The multi-group code APOLLO2-MORET 4 using the JEF2.2 library,
- the multi-group code APOLLO2-MORET 5 using the JEFF-3.1.1 library,
- the continuous energy MORET 5.D.1 code [3] using the JEFF-3.3 library.

APOLLO2 [10] is a one-dimensional lattice code used for the preparation of multi-group cross-sections in equivalent cell approximation. The cross-sections and fluxes are described with a 172-group (APOLLO2-MORET 4 route) or 281-group (APOLLO2-MORET 5 route) structure based on cross-sections coming from the CEA93 V6 (APOLLO2-MORET 4 route) or CEAV5.1.2 (APOLLO2-MORET 5 route) library. All isotopes are developed in P1 Legendre polynomials, except moderating elements (such as $H_2O$) and heavy nuclides (U, Pu), which are developed in the 5th and 9th orders, respectively.

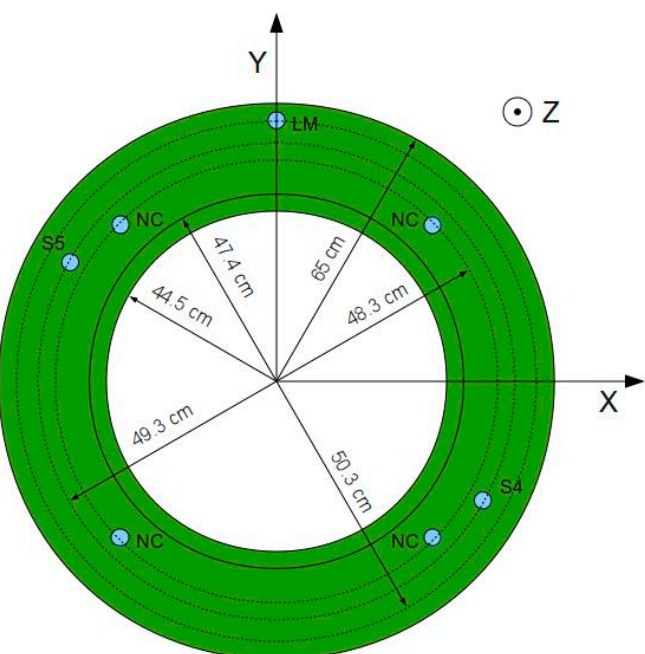

**Figure 8.** Position of neutron counters (NC).

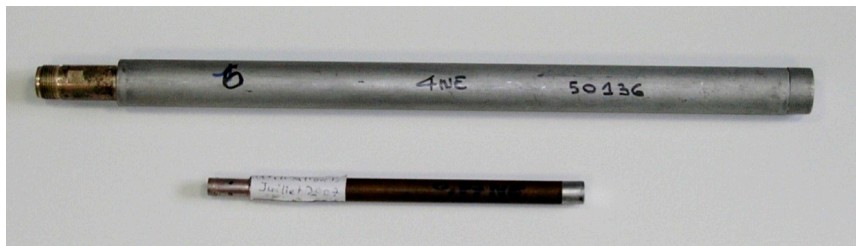

**Figure 9.** Neutron counters with their holders.

Cross-sections for APOLLO 2 in the APOLLO2-MORET 5 route are interpolated in temperatures.

MORET 5.D.1 is a 3D Monte Carlo code for the neutron transport calculation. It can use either:

- Macroscopic homogenized, self-shielded cross-sections generated by the APOLLO2 code for the multi-group mode,
- continuous energy cross-sections processed in the ACE format by the IRSN GAIA1 tool [11] based on NJOY2016.35 [2] for the continuous energy mode.

In general, MORET 5.D.1 uses the cross-sections generated at pre-defined temperatures and makes the 3D calculation using the closest available temperature. For the purpose of this study, cross-sections at 295.15 K and 301.15 K were generated using the in-house IRSN GAIA tool for all the free gas cross-sections. For the thermal scattering data of hydrogen in water, JEFF-3.3, ENDF/B-VIII.0, or a new evaluation based on experimental data were used.

### 5.3. Methodology to Interpret the Experimental Results

Using the information provided in Table 1, the critical mass of plutonium can be determined for each experiment, whatever the temperature. The uncertainty associated with the plutonium mass is calculated using the following Formula (2):

$$\sigma_{mass} = \sqrt{\left(\left(\frac{\partial mass}{\partial C(Pu)}\right)^2 \times \left(\sigma_{C(Pu)}{}^2\right)\right) + \left(\left(\frac{\partial mass}{\partial C(R)}\right)^2 \times (\sigma_R{}^2)\right) + \left(\left(\frac{\partial mass}{\partial Hc}\right)^2 \times (\sigma_{Hc}{}^2)\right)} \qquad (2)$$

where $m_{Pu} = C(Pu) \times \pi \times R^2 \times H_c$, R is the radius of the solution tank, and Hc is the critical solution height.

The critical plutonium mass versus temperature is reported for each experiment per type of plutonium concentration in Figures 10–12.

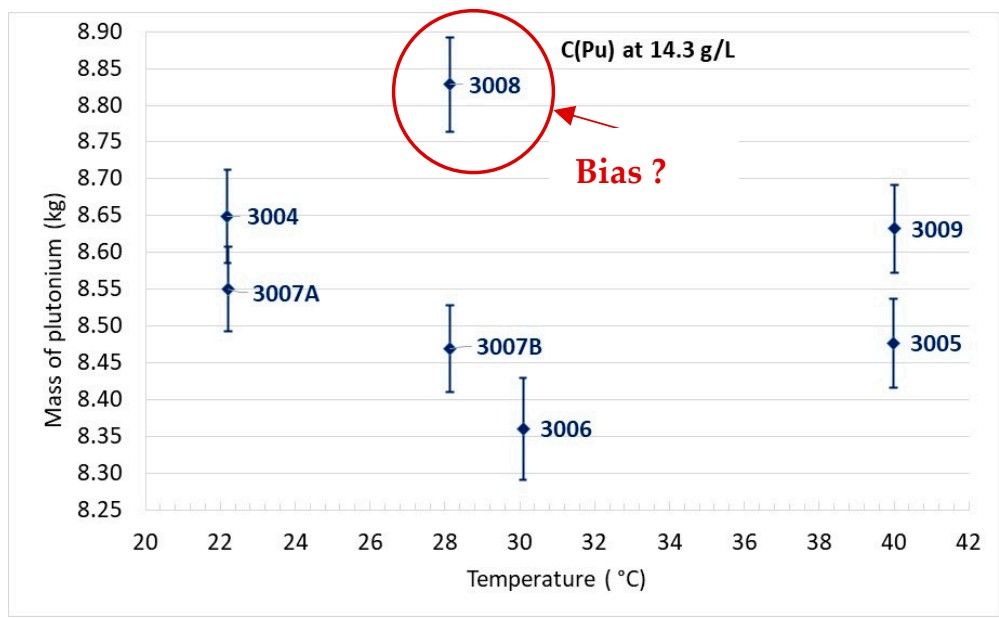

**Figure 10.** Critical mass of plutonium for C(Pu) = 14.3 g/L.

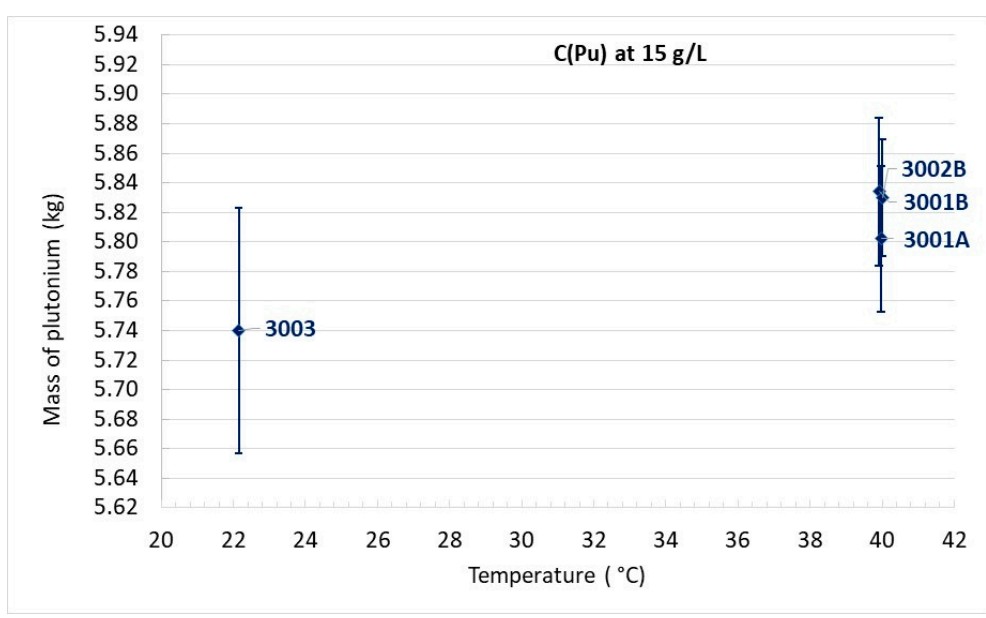

**Figure 11.** Critical mass of plutonium for C(Pu) = 15 g/L.

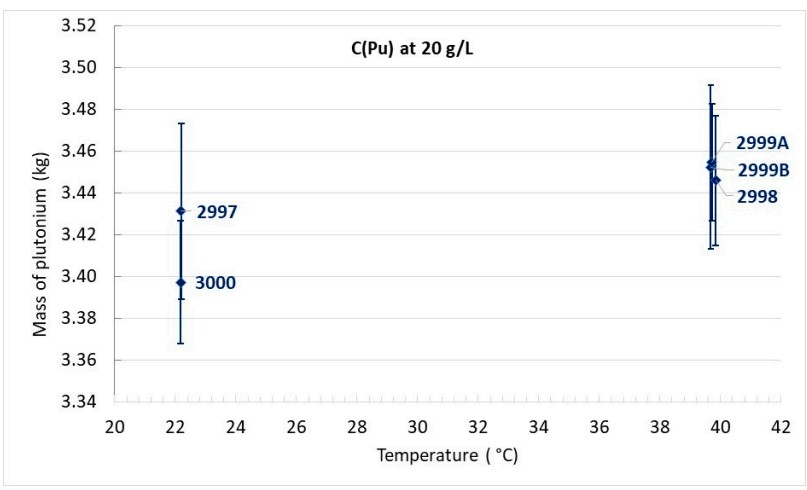

**Figure 12.** Critical mass of plutonium for C(Pu) = 20 g/L.

For plutonium concentrations close to 15 g/L or 20 g/L, it is difficult to conclude the existence of a temperature effect since the discrepancy in plutonium mass comprises the uncertainty margins.

For plutonium concentrations close to 14.3 g/L, it can be pointed out that for experiments performed at the same temperature and with quite an identical plutonium concentration, a discrepancy in the plutonium mass comprised between 0.15 kg and 0.4 kg can be observed. This results in a difference in critical height comprised between 2 cm (3004/3007A) and 5.8 cm (3007B/3008) (see Table 1). Consequently, it can be inferred that the uncertainty of experiment reproducibility is significant. With the strong difference in the critical height and the derived critical mass between experiments 3007B and 3008, one cannot exclude a bias in experiment 3008 that is inconsistent with experiments performed at the same plutonium concentration and at the same temperature.

When looking at the same figures, it appears that the PU+ effect is lower than this uncertainty; independent experiments are therefore of low value when trying to highlight the effect. It is, therefore, necessary to magnify the reactivity effect while minimizing the experimental uncertainties.

As mentioned in Section 3.1, one way to do this is to perform two sub-critical approaches with the same solution. This was performed for experiments 3007A and 3007B. In that case, the only uncertainty in the experiment comes from the critical height measurement, resulting in uncertainty on the critical mass equal to 2 g.

The discrepancy of 81 g in the plutonium mass between the experiment performed at 22 °C and that performed at 28 °C is consequently significant compared to the experimental uncertainties.

### 5.4. Calculation of the Temperature Effect

The main goal of the plutonium temperature effect program was to exhibit the effect of temperature on the plutonium solution density experimentally. The experimental results followed the results published by Toshihiro et al., where a positive temperature coefficient was observed. One of the aims of this paper is to confirm the above positive effect by performing neutronic simulations.

#### 5.4.1. Preparation of Input Data for Calculation

For the 14.3 g/L plutonium concentration encountered in the program, the effect of the temperature variation was quantified with the multi-group APOLLO2-MORET 4 code using the JEF2.2 library, the multi-group APOLLO2-MORET 5 code using the JEFF-3.1.1 library, and the continuous energy Monte Carlo code MORET 5.D.1 using the JEFF-33 library. Two input decks were generated for experiments 3007A and 3007B, corresponding to the 14.3 g/L concentration. One corresponds with the experiment at room temperature.

The other refers to the same experiment but at a temperature of 28 °C (3007A). The mass of plutonium is constant; the height of the solution is the only variable. The dilation factor is equal to $\frac{C(Pu)_T}{C(Pu)_{21\,°C}}$. This factor has been determined using experimental data and a density formula based on the Kumar and Koganti theory [12].

5.4.2. Generation of Thermal Scattering for Continuous Energy Codes

Among the three effects responsible for the PU+ effect, the thermal scattering effect of hydrogen in the water molecules is the most difficult to quantify. Indeed, this effect, caused by the influence of chemical bindings and atom arrangements in the crystalline structure of water molecules, impacts the thermalization of neutrons. The treatment of the thermal scattering of hydrogen is done directly using S($\alpha$,$\beta$), often referred to as the thermal scattering law (TSL), which defines the modified scattering cross-sections and angular distributions in the thermal energy range.

For experiments conducted at 28 °C or 30 °C, TSL data in JEFF-3.3 were only available at 21 °C or 50 °C, which was not sufficient if one hopes to exhibit such a small reactivity effect. Consequently, various efforts were made to interpolate S($\alpha$,$\beta$) thermal scattering data at the exact experimental temperatures. These works are detailed in previous papers [5,6]. Three different ways of dealing with the problem were investigated:

- A temperature interpolation between the different S($\alpha$,$\beta$) tables as they are given in their basic ENDF evaluation;
- interpolation between processed S($\alpha$,$\beta$) tables (beginning with a temperature);
- interpolation between the modified cross-sections for hydrogen.

Available TSL Data in Standard Nuclear Data Evaluations

TSL data for light water available in the standard nuclear data libraries were investigated. It is known that the TSL data available to the users in the evaluated ENDF files are for a fixed grid of temperatures. Users often make an approximation for TSL data at required temperatures, such as the closest temperature approach or interpolation of the cross-sections. Various other methods are presently available to users depending on the Monte Carlo codes, such as Serpent, MONK, MCNP, and OpenMC. Stochastic mixing is one such method that helps users treat the problem of using TSLs at required temperatures by mixing two TSLs at nearby temperatures stochastically. The choice of these options may lead to erroneous results at times for temperatures close to the midpoints of the two closest temperatures or for benchmarks that are very sensitive to TSL temperatures.

The Monte Carlo code chosen in this study, MORET 5.D.1, only has the option of using the closest temperature available in the TSL processed file. In the former case, the temperatures close to 22 °C and 28 °C in the JEFF-3.3 evaluation are 293.6 K. In the case of ENDF/B-VIII.0, the two closest temperatures that can be used are 293.6 K (for a 22 °C case) and 300 K (for a 28 °C case). These two data libraries were used to test the temperature effect using the closest temperature approach.

Using the closest temperature approach for studying the plutonium temperature effect might not be the most appropriate solution. One may need to regenerate new TSLs at the required temperatures either by performing new time-of-flight experiments or by interpolating the LEAPR (LEAPR is a module of NJOY that calculates the thermal neutron scattering laws for use in reactor physics calculations) input parameters for the required temperature. This approach is explained in the next section.

New TSL Evaluation for Light Water Based on Recent Time of Flight (TOF) Experimental Data

IRSN is working on the development of improved TSL evaluation for light water based on recent high-resolution time-of-flight inelastic neutron scattering (INS) measurements. This work aims to generate new TSL libraries for light water not only for room temperatures but also for high temperatures close to reactor operating temperatures. In particular, the primary objective was to study the temperature dependence of the TSL. In addition, the INS (inelastic neutron scattering) experiments were carried out at the Spallation neutron source (SNS) at the

Oak Ridge National Laboratory, United States, for a series of temperatures, pressures, and incident neutron energies. Details about the experiment can be found in reference [13]. The temperature of interest for this work is the measured data from SNS for 295 K and 323.6 K. The experimentally measured phonon spectrum was used to generate the TSL for light water at 22 °C and 28 °C using the SAB module of GAIA [14]. The SAB module interpolates the necessary LEAPR input parameters and the phonon spectrum at the required temperature and provides an ENDF file that GAIA processes using NJOY 2016.35. The plutonium temperature experimental program is also an opportunity to test the newly developed TSLs and verify if a positive effect is observed using these new cross-sections.

*5.5. Calculated Temperature Coefficients*

Only the 3007A configuration was considered since it is the only one where the initial and perturbed cases are strongly correlated, with the approach-to-critical for the 28 °C configuration (3007B) being performed in one run after draining the water level and increasing the temperature of the plutonium nitrate solution. The contributions of the Doppler, thermal expansion, and thermal scattering effects were evaluated.

The temperature coefficients were calculated using the multi-group codes APOLLO2-MORET 4 and APOLLO2-MORET 5 using, respectively, nuclear data based on the JEF2.2 and JEFF-3.1.1 evaluations and also the continuous energy code MORET 5.D.1. For MORET 5.D.1, the JEFF-3.3 library was employed for all nuclides except hydrogen in water, where thermal scattering data from various evaluations or by regenerating at correct temperatures based on new experimental data were used.

The geometry is assumed not to vary between 22 °C and 28 °C. The only discrepancy between the two configurations is the composition of the plutonium and water reflectors after the expansion solution (when the expansion effect is accounted for), their temperatures, and the thermal scattering data of hydrogen in water.

A first analysis of Table 5 shows that positive temperature coefficients are obtained for the multi-group APOLLO2-MORET 4 and APOLLO2-MORET 5 codes. Indeed, for these codes, the thermal scattering data of hydrogen are calculated at a tabulated temperature if the temperature of the benchmark is less than 2 °C from the tabulated temperature; otherwise, an interpolation at the benchmark temperature is performed.

**Table 5.** Temperature coefficients (pcm/K) calculated with CRISTAL package codes (multi-group) and libraries.

| Experiment | C(Pu) in g/L | APOLLO2-MORET 4 | APOLLO2-MORET 5 |
|---|---|---|---|
| | | JEF2.2 | JEFF-3.1.1 |
| 3007A/B | 14.294 | $15.5 \pm 2.4$ | $16.2 \pm 2.4$ |

However, it can be seen in Table 5 that the obtained temperature coefficient is significantly higher than the experimentally measured value (5.17 pcm/K, as shown in Table 6, last column). One of the reasons that can be attributed to this large discrepancy is the interpolation of thermal scattering cross-sections by APOLLO2 between two broad temperature grids available in the JEFF-2.2 and JEFF 3.1.1 TSL evaluations for light water, i.e., 293.6 K and 350.0 K.

**Table 6.** Effect of temperature on the critical height and translation in reactivity worth (SNS).

| Experiment | C(Pu) in g/L | Critical Height (cm) | Difference in the Critical Height (mm) | Reactivity Worth of 1 mm of Solution (pcm) | Reactivity Worth Corresponding to the Temperature Effect (pcm) | Reactivity Worth Corresponding to the Temperature Effect (pcm/K) |
|---|---|---|---|---|---|---|
| 3007A | 14.294 | 133.254 | −12.57 | 2.47 | 31 | 5.17 |

The same calculation was performed using the MORET 5.D.1 continuous energy codes with nuclear data based on JEFF-3.3 processed for the temperatures encountered in the PU+ program (28 °C), the thermal scattering data being obtained from JEFF-3.3, ENDF/B-VIII.0 at the temperatures available in the ENDF files (293.15 K and 300 K for ENDF/B-VIII.0 and 293.15 K for JEFF-3.3), and the newly developed TSL based on the SNS data at the required temperatures (295.15 K and 301.15 K). The temperature coefficients obtained are reported in Table 7.

**Table 7.** Temperature coefficients (pcm/K) calculated with MORET 5 (continuous energy) and libraries (expansion effect of solution taken into account)—Monte Carlo standard deviation = 0.0005.

| C(Pu) in g/L(3007 A) | Temperature Effect | TSL from JEFF-3.3 at 293.6 K and 293.6 K | TSL from ENDF/B-VIII.0 at 293.15 K and 300 K | TSL with New Evaluation from SNS [11] at 295.15 K and 301.15 K |
|---|---|---|---|---|
| | | MORET 5 | | |
| | Thermal expansion | +0.33 | +0.33 | +0.33 |
| 14.294 | Doppler | +3.67 | +3.67 | +3.67 |
| | $S(\alpha,\beta)$ | 0 | +7.17 | +5.83 |
| | Total | +3.33 | +9.67 | +11.83 |

As expected, when referring to the experimental results, a positive temperature coefficient is obtained (see Table 7) when using the TSL data from JEFF-3.3; only a +3.33 pcm/K effect is observed as the same TSL data at 293.6 K was utilized for both the simulations at 22 °C and 28 °C. The effect is larger when using TSL data from ENDF/B-VIII.0 (9.67 pcm/K) and from SNS (+11.83 pcm/K). Considering their uncertainties, these values are consistent with the reactivity worth of the solution level decreasing due to the temperature effect (+5.17 pcm/K in Table 7).

Another conclusion is that the main contributor to the temperature effect is the thermal scattering data effect, followed by the Doppler effect. The expansion effect is calculated to be negligible.

Indeed, when looking only at the effect of TSL data without considering the expansion effect of the Pu solution, it appears that the effect of TSL data is negligible for TSL data from JEFF-3.3 but larger for TSL data from ENDF/B-VIII.0 (+7.17 pcm/K) and from SNS (+5.83 pcm/K).

This observed negligible effect of TSL data for JEFF-3.3 is evident due to the unavailability of TSL data for a fine temperature grid in the evaluation. A close observation of the larger positive temperature coefficient obtained using ENDF/B-VIII.0 and the new TSL data highlights the importance of having either a fine grid in TSL temperatures in the nuclear data evaluation or regeneration of TSLs at the required temperature for benchmarks sensitive to TSL temperatures.

## 6. Conclusions

The PU+ theoretical effect was already studied by Yamamoto and Myioshi in 2002. They showed that such an effect could be predicted for lowly concentrated plutonium solutions. The effect is the result of a competition between three main physical parameters: Density of the solution, Doppler, and moderation effects. The authors showed that the burnup contributed to reinforcing the temperature effect through an increase in [241]Pu in the plutonium isotopic vector and that this effect is emphasized with the decay of [241]Pu in [241]Am.

However, there was no evidence of the effect until 2007. The PU+ program conducted at the CEA Valduc research center from 2006 to 2007 supported this theory. Two sets of experiments were performed: One comprising independent sub-critical approaches and the other involving two correlated approaches (sub-critical approaches with partial draining of the plutonium solution and heating the solution through the water reflector). Finally, only the two correlated approaches allowed for a positive PU+ effect. Indeed, a lower plutonium mass was obtained for the experiment at 28 °C (3007 B) than for the experiment

at 21 °C (3007A), and the discrepancy proved to be significant concerning the experimental uncertainties. The calculations confirmed this experimental result with the multi-group APOLLO2-MORET 4 and APOLLO2-MORET 5 codes. However, thermal scattering data at the correct temperatures were needed to point out the effect with continuous energy codes from Monte Carlo, such as MORET 5. Two simulations were carried out, one for experiment 3007A, corresponding to 21 °C, and the other for experiment 3007B, corresponding to 28 °C. Several TSL data sets available in JEFF-3.3 and ENDF/B-VIII.0, as well as a new TSL evaluation for light water based on recent experimental data from SNS, were tested to study the impact of TSL on this benchmark and the experimentally observed positive effect. It was observed that the TSL data present in JEFF-3.3 at the closest temperatures to our experiment showed a negligible positive effect. In contrast, the effect was larger with ENDF/B-VIII.0 and the new TSL evaluation. Further work is needed to model the positive effect with better accuracy by developing and incorporating the TSL data for Pu in PuN solutions that may have an impact on the simulation results.

**Author Contributions:** Conceptualization, N.L.; methodology, V.J. All authors have read and agreed to the published version of the manuscript.

**Funding:** This research received no external funding.

**Data Availability Statement:** Data are not available due to privacy restrictions.

**Acknowledgments:** The authors wish to thank Mike Zerkle (Bettis Atomic Power Laboratory) for providing his valuable feedback in relation to the generation of TSL libraries by interpolating the LEAPR input parameters; they would also like to thank Pascal Grivot, who, as an experimentalist, provided feedback on the way the experiments were conducted and finally, a special thanks to Benoît Normand, who was at the origin of the first evaluation of the experiments, and also to Wim Haeck (Los Alamos National Laboratory, formerly IRSN), who made the evaluation of the experiments for the ICSBEP working group.

**Conflicts of Interest:** The authors declare no conflict of interest.

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
