# Peer review of "The Plutonium Temperature Effect Program"

_jne, doi:10.3390/jne4030035_

Round 1
Reviewer 1 Report
The paper provides initial confirmation of a positive temperature effect in diluted plutonium solutions through an experimental program. The overall design of the experiment is well-planned, and the authors have made efforts to minimize experimental uncertainties, which can be beneficial for other researchers in the field. The obtained results support the existence of a positive temperature effect. However, one limitation worth noting is that the calculation relies on only one pair of data, which may not provide a sufficiently robust foundation. Despite this limitation, the paper remains well-aligned with the scope of the journal, and I would recommend its consideration with some minor modifications.
Specific comments (MA = major, MI = minor, OP = optional)
1. [MA] Abstract: Could you please provide a concise summary highlighting the significance of the paper at the end of the abstract?
2. [MI] P1, L28. Consider adding a reference.
3. [MA] Figure 1. Could you please provide a figure with a higher resolution? The current resolution makes it difficult to zoom in and read the text clearly.
4. [MI] P2, L66. Could you please format the citation for “reference4” in a similar manner to the other citations?
5. [MA] P3, L91. I cannot find “205” cm in Figure 2. Could you please clarify?
6. [MI] P3, L98-99. Could you please label “120.6 cm”, “232 cm”, and “rock wool” in Figures?
7. [MA] Figure 4. I noticed that you did not mention it in the text between Figure 3 and Figure 5.
8. [MA] Figure 7. Again, could you please provide a figure with a higher resolution?
9. [MI] P6, L157. How were concentration and temperature values determined?
10. [MI] P7, L178. “For some experiments” I would recommend replacing “some” with “these” since you are specifically referring to the four experiments that were terminated prematurely.
11. [MA] Table I. I recommend including the phase I/II labels and the terminated experiments in the table. I cannot locate five experiments at 20 g/l, four experiments at 15 g/l, and five experiments at 14.3 g/l for phase I, as well as three experiments at 14.3 g/l for phase II.
12. [MI] P9, L210. Could you provide a rough estimate or quantify the phrase "increased slowly by"?
13. [MI] P11, L292. Same here. Could you quantify the “detection limit”?
14. [MI] P11, L297. I believe this line should follow the same format as lines 292-296.
15. [MA] Figure 8. Again, could you please provide a figure with a higher resolution?
16. [OP] P12, L317. Any reference for MORET 5.D.1?
17. [MI] P13, L348. Do you have any insights or speculation about the possible source of the bias?
18. [MI] Figures 11 & 12. There are no labels in the two figures.
19. [MA] P15, L379. While the 3007A and 3007B data were obtained from an improved experiment, have you conducted any further experiments to validate their reliability?
20. [MI] P16, L429. “TOF” Please provide the complete term for this abbreviation. I believe it first appeared in line 425.
21. [MI] P18, L534. Are there any plans to generate additional data samples in order to validate whether the reported temperature coefficient is repeatable?
The paper exhibits a clear organizational structure and employs fluent language.
Author Response
Word document submitted

Reviewer 2 Report
This paper provides the results of experimental measurements of temperature effects in plutonium solutions that demonstrate a positive temperature reactivity coefficient in the solution at a concentration of approximately 14.3 g/l Pu in solution for a vector of 77% Pu-239, 21% Pu-240, and around 1% or less of remaining isotopes. It also provides an assessment of numerical simulations of these experiments, investigating TSL data for different libraries with different versions of the MORET Monte Carlo code.
The paper is very complete, well written and organized. I do feel that inclusion of the Phase I measurements, which ultimately have little value, do introduce some confusion at first as their relevance wasn’t initially clear. However, I do feel that inclusion of this information is important in a lessons learned sense. I do think it lacks a discussion of the significance of this finding (other than confirming Yamamoto) and as noted below some indication of what causes this behavior.
General thoughts I had were:
1) the paper does not describe the mechanism that causes this reactivity behavior. What did Yamamoto use to predict this behavior? The TSL study in the paper seems to indicate that it is a result in scattering in bound hydrogen, but its relationship to plutonium in solution is not clear. Nor are the TSL measurements made in the presence of plutonium. It has to be a relationship between hydrogen scattering and plutonium, right?
2) Relating to (1), not knowing what the mechanism is, would the same behavior be noted for a different plutonium vector? Would a positive reactivity coefficient be seen in a plutonium vector coming from recycled fuel? Perhaps discuss this in the conclusions?
3) Similarly would the magnitude of the reactivity coefficient change as a function of concentration? What does Yamamoto predict?
4) Perhaps Phase 1 and phase 2 discussions should be completely separated, including introduction of the data. As noted earlier this did cause some confusion. At first I thought Phase 2 was just intended to augment the missing data from Phase 1, but it later became clear that the tank configuration and measurement process were changed.
Reading through the manuscript I have some comments and questions:
Around line 89, the statement “…provides neutron reflection by a water layer of 22 cm laterally and under the plutonium vessel…” could be rewritten as “…provides neutron reflection by a water layer of 22 cm both laterally and under the plutonium vessel…” to make it clear that the bottom layer is also 22 cm. This is of course shown in Fig. 3.
In Figure 3, it is not clear what Z2 CND 18-10 and …17-12 are. After a Google search I now understand. Maybe add “stainless steel” after the name in the legend or define them as SS in the text.
In lines 164-166, I am not familiar with the 1-Beff/10 approach, but as I think about this it makes sense as a safety approach. Is there a reference for this approach, and where does the factor of 10 come from? Maybe a little more description of this approach would be of value for the reader unfamiliar with it?
Lines 174-175, why were concentration values of 14.3, 15 and 20 selected? 10, 15, and 20 I would understand, but why the specific value of 14.3? Would it have been significantly different that 15?
Table 1 and its description, it would be informative why a series A and B were done for some experiments - what was common to experiment 3001A and 3001B that distinguishes them from 3000? Can this be described or is in just a common internal nomenclature with no real significance for this measurements?
Line 190, you mention “two separate approaches” it is not clear to me what two separate approaches you mean.
Lines 207-209 you indicate that ~1% of the initial solution would have to be drained, which I understand and accept. However, in the next bullet point in lines 210-212, you discuss a subcritical approach at the higher temperature, which means you would have had to remove more than 1% of the volume to be able to do the approach to critical. So how does the 1% figure in?
It sounds like only the liquid in the main tank was heated, not the fill tank. So when you added more liquid during the approach to critical, did you have to wait for the added liquid to come to temperature?
Lines 313-314, sentence starting with “All isotopes are developed in P1 Legendre polynomials…” I am not familiar with using polynomials in multi group libraries. Can you provide more information on what you are doing with the polynomials of various orders?
Line 352-353, “...getting rid of the experimental uncertainties.” I don’t think you can get rid of them, only minimize them.
Line 412, “SERPENT” should be spelled “Serpent.” It is not an acronym and is usually written with only the first letter capitalized.
Line 426, “LEAPR” - the casual reader would not know what LEAPR is and what it does. Please provide a little background.
Line 436, please define INS. Also, consider adding “In addition,” to the beginning of the sentence so that it is clear that this is not related to the IRSN experiments.
Line 473, you compare to the experimentally measured value, but you have not provided that value yet. You should provide the experimentally measured value and point to Table 7.
Line 508 you mention that nuclear data evaluations be performed on a fine grid in TSL temperatures to help resolve this issue. Doing such detail would require an immense amount of effort and cost, and would make the evaluation data files even bigger. Beyond the interest to support this very specific experiment, surely there is little added value to such fine grid measurements (and associated theoretical fits to the measurements). Or am I not understanding what you are suggesting?
I'm a little concerned about the significant range in TSL evaluation results. It would be good if this could be tied to the expected phenomena.
The quality of English in this manuscript is very good and only very minor editing would be needed, mostly in word usage, not in grammar (e.g., "till" should be "until"). There was no difficulty in reading the text. There was some ambiguity in the discussion as discussed earlier, but I don't believe this is a language issue.
